# CONDITIONED INITIALIZATION FOR ATTENTION

**Hemanth Saratchandran**
Australian Institute for Machine Learning
Adelaide University

**Simon Lucey**
Australian Institute for Machine Learning
Adelaide University

## ABSTRACT

Transformers are a dominant architecture in modern machine learning, powering applications across vision, language, and beyond. At the core of their success lies the attention layer, where the query, key, and value matrices determine how token dependencies are captured. While considerable work has focused on scaling and optimizing Transformers, comparatively little attention has been paid to how the weights of the queries, keys and values are initialized. Common practice relies on random initialization or alternatives such as mimetic initialization, which imitates weight patterns from converged models, and weight selection, which transfers weights from a teacher model. In this paper, we argue that initialization can introduce an optimization bias that fundamentally shapes training dynamics. We propose **conditioned initialization**, a principled scheme that initializes attention weights to improve the spectral properties of the attention layer. Theoretically, we show that conditioned initialization can potentially reduce the condition number of the attention Jacobian, leading to more stable optimization. Empirically, it accelerates convergence and improves generalization across diverse applications, highlighting conditioning as a critical yet underexplored area for advancing Transformer performance. Importantly, conditioned initialization is simple to apply and integrates seamlessly into a wide range of Transformer architectures.

## 1 INTRODUCTION

Transformers (Vaswani et al., 2017) have rapidly become a cornerstone of modern machine learning, driving progress in fields as diverse as natural language processing (Vaswani et al., 2017; Zhuang et al., 2021; Zhen et al., 2022), computer vision (Dosovitskiy et al., 2020; Liu et al., 2021; Touvron et al., 2021; Carion et al., 2020), and robotics (Salzmann et al., 2020; Maiti et al., 2023). A key factor behind this versatility is the self-attention mechanism, which models interactions between tokens by comparing them pairwise and dynamically weighting their contributions. This ability to capture both long-range and global dependencies has positioned Transformers as a foundational architecture across a wide spectrum of learning tasks.

While substantial effort has been devoted to improving the efficiency, scalability, and expressiveness of attention mechanisms (Ali et al., 2021; Xiong et al., 2021b; Ding et al., 2022), relatively little focus has been placed on a more basic but equally important aspect: *how attention weights are initialized*. Initialization plays a critical role in shaping optimization landscapes of deep neural networks. Classical schemes such as Xavier (Glorot & Bengio, 2010) and Kaiming (He et al., 2015) initialization showed that carefully chosen scaling of weights at the start of training can dramatically improve gradient optimization and stability in deep networks. These insights were crucial for enabling the training of very deep architectures such as ResNets (He et al., 2016), where poor initialization could otherwise cause vanishing or exploding gradients. Yet, despite their depth and complexity, Transformers have received far less theoretical scrutiny on this front. Given that self-attention relies on query, key, and value projections whose interplay directly governs the stability of token interactions, it is natural to ask whether initialization schemes tailored specifically to attention could offer similar benefits.

Currently, Transformers typically adopt simple random initializations, without consideration of the unique structure of attention layers. Recent alternatives such as *mimetic initialization* (Trockman & Kolter, 2023), which transfers statistical patterns from converged models, and *weight selection* (Xu et al., 2023), which reuses pretrained weights from larger teacher models, highlight a growing recog-

nition that initialization matters. However, these methods remain heuristic and lack a principled connection to the conditioning of the attention mechanism itself.

In this work, we revisit Transformer initialization from a theoretical perspective. We show that the stability of optimization in self-attention layers is closely tied to the conditioning of their Jacobians, which in turn depends on the spectral properties of the query, key, and value projections. Building on this insight, we introduce **conditioned initialization**, a principled scheme designed to improve the spectral conditioning of attention blocks at the start of training. Rather than directly modifying training objectives, our method intervenes at initialization, providing an inductive bias that promotes more stable optimization dynamics.

Our contributions are threefold:

1. **Theoretical framework:** We establish a connection between the conditioning of self-attention Jacobians and the spectral structure of the query, key, and value matrices, motivating initialization schemes that explicitly target this property.

2. **Conditioned initialization:** We propose a simple initialization method that reduces the upper bound on the attention Jacobian's condition number, thereby biasing training toward stable optimization.

3. **Empirical validation:** Through experiments on diverse benchmarks, spanning image classification, object detection, instance segmentation, language modeling, and long-range sequence learning, we show that conditioned initialization consistently accelerates convergence and improves generalization, and can be easily integrated into a variety of different Transformer architectures.

By highlighting initialization as a critical yet underexplored component of Transformer design, our work opens up a new perspective on how principled conditioning can be leveraged to improve optimization stability and downstream performance.

## 2 RELATED WORK

**Initialization.** Initialization strategies play a pivotal role in determining how efficiently deep networks can be trained. Early advances such as Xavier (Glorot & Bengio, 2010) and Kaiming initialization (He et al., 2015) demonstrated that properly scaling weights at the outset helps preserve variance across layers, preventing issues like vanishing or exploding gradients. These ideas were foundational in enabling the successful training of very deep architectures, most notably ResNets (He et al., 2016). In the context of Transformers, however, initialization has typically been treated in a more ad hoc manner, with standard practice relying on simple schemes from normal or truncated normal distributions. More recent efforts have begun to acknowledge the unique structure of attention layers. For example, mimetic initialization (Trockman & Kolter, 2023) introduces inductive bias by imitating the statistical patterns of trained networks, while weight selection (Xu et al., 2023) transfers weights from larger pretrained teacher models to provide a stronger starting point for smaller architectures. In our experiments we compare to these two methods.

**Conditioning.** A growing body of work has underscored the importance of conditioning for both the trainability and generalization of neural networks. In particular, Saratchandran et al. (2025) showed that networks with better-conditioned weights tend to achieve superior performance, and proposed a matrix preconditioning method to explicitly control condition numbers during training. From a different angle, Liu et al. (2022) analyzed optimization through the lens of the neural tangent kernel (NTK), demonstrating that well-conditioned NTKs lead to faster and more reliable convergence, particularly in the infinite-width regime where the NTK dominates learning dynamics (Jacot et al., 2018). This has been extended to the transformer setting in Yang (2020). Other studies have pointed out structural factors that affect conditioning. For example, Agarwal et al. (2021) observed that increasing network depth can itself improve conditioning, thereby aiding gradient-based methods. In the case of Transformers, Ji et al. (2025) argued that skip connections act as an implicit conditioning mechanism, stabilizing the optimization of deep Transformers. Our work departs from these approaches by asking whether initialization itself can be designed to yield better-conditioned attention layers from the outset.

## 3 THEORETICAL FRAMEWORK

### 3.1 PRELIMINARIES

For the theoretical framework we will primarily focus on self-attention, which is one of the most common forms of attention in a Transformer. Self-attention is composed of three learnable matrices, query $W_Q \in \mathbb{R}^{D \times d}$, key $W_K \in \mathbb{R}^{D \times d}$, and value $W_V \in \mathbb{R}^{D \times d}$ defined for an input sequence $X \in \mathbb{R}^{N \times D}$.

$$\mathrm{A}(X) = \mathrm{softmax}(X W_Q W_K^T X^T) X W_V \tag{1}$$

where $\mathrm{softmax}$ is the softmax activation that acts row-wise on a matrix (Prince, 2023). Note that then $\mathrm{A}(X) \in \mathbb{R}^{N \times d}$. In general, Transformers employ multiple heads i.e. attention matrices $\mathrm{A}_i$ for $1 \leq i \leq h$ where $h$ is the number of heads. These are then concatenated together to form a multi-head attention layer $[A_1, \ldots, A_h]$. We point out that in some references the notation $\mathrm{A}(X)$ is reserved for only the term $\mathrm{softmax}(X W_Q W_K^T X^T)$. However, in this paper as we will be concerned with the whole attention layer we use $\mathrm{A}(X)$ to denote the whole layer output as defined in eq. (1). For further details on Transformers readers may consult Prince (2023).

The self-attention map of a layer in a Transformer $\mathrm{A}(X)$ has parameters given by those parameters in $X$ from the previous layer and those given by $W_Q$, $W_K$, $W_V$ that define $\mathrm{A}(X)$. Our work will consider the Jacobian of $\mathrm{A}(X)$ with respect to the parameters within the layer of $\mathrm{A}(X)$, namely $W_Q$, $W_K$, $W_V$. Therefore, when we speak of the Jacobian of $\mathrm{A}(X)$ it will be with respect to $W_Q$, $W_K$, $W_V$. We will denote this Jacobian by $\mathrm{J}(\mathrm{A}(X))$ and note that it is defined by

$$\mathrm{J}(\mathrm{A}(X)) = \left[ \frac{\partial \mathrm{A}(X)}{\partial W_Q}, \frac{\partial \mathrm{A}(X)}{\partial W_K}, \frac{\partial \mathrm{A}(X)}{\partial W_V} \right]^T \tag{2}$$

Given a matrix $Z \in \mathbb{R}^{m \times n}$ we denote the vectorization of $Z$ by $\mathrm{vec}(Z) \in \mathbb{R}^{mn \times 1}$ (Magnus & Neudecker, 2019). Note that for such a matrix there is a transformation $T_{mn} \in \mathbb{R}^{mn \times mn}$ such that $T_{mn} \mathrm{vec}(Z) = \mathrm{vec}(Z^T)$ where $Z^T$ denotes the transpose of $Z$. The matrix $T_{mn}$ is known as a commutation matrix and is a permutation matrix (Magnus & Neudecker, 2019). The maximum singular value of a matrix $Z$ will be denoted by $\sigma_{\max}(Z)$ and the minimum singular value by $\sigma_{\min}(Z)$. We will use the standard terminology SVD to denote the singular value decomposition of a matrix. Given a vector $v \in \mathbb{R}^n$ the notation $||v||_2$ will denote the vector 2-norm of $v$. Finally, we will let $I_{m \times n}$ denote the rectangular identity matrix that has all 1's on its main diagonal and $\mathcal{O}_{m \times n}$ as the real $m \times n$ semi-orthogonal matrices.

### 3.2 MAIN THEOREMS

In this section, we present the main theorem of the paper, which motivates the development of a simple yet effective strategy, conditioned initialization, designed to reduce the condition number of the Jacobian of the attention layer at initialization. As shown in section 4, this scheme is straightforward to implement while providing significant optimization benefits.

**Definition 3.1.** Let $Z$ be an $N \times d$ matrix of full rank. The condition number of $Z$, denoted by $\kappa$, is defined as

$$\kappa(Z) = \frac{\sigma_{\max}(Z)}{\sigma_{\min}(Z)} \tag{3}$$

where $\sigma_{\max}(Z)$ denotes the maximum singular value of $Z$ and $\sigma_{\min}(Z)$ the minimum singular value of $Z$, which we know is non-zero as $Z$ is full rank.

Our objective is to analyze the condition number of the self-attention layer in a Transformer. We show that the condition number of its Jacobian depends on the condition numbers of the query, key, and value weight matrices. Furthermore, we demonstrate that initializing $W_Q$, $W_K$, and $W_V$ with low condition numbers imparts an inductive bias into the Transformer architecture that leads to more effective optimization.

We begin by examining the derivatives of the self-attention layer with respect to the parameters $W_Q$, $W_K$, and $W_V$. We will need the following lemma.

**Lemma 3.1.** *Let $\Lambda : \mathbb{R}^n \to \mathbb{R}^{n \times n}$ denote the function $\Lambda(z) = Diag(z) - z \cdot z^T$. We then have that*

$$\frac{\partial \text{softmax}}{\partial x}(z) = \Lambda(\text{softmax}(z)). \tag{4}$$

**Proposition 3.1.** *Let $\mathrm{A}(X)$ denote a self-attention matrix with input $X$ as defined by eq.* (1). *Then*

$$\frac{\partial \mathrm{A}(X)}{\partial W_Q} = (W_V^T X^T \otimes I_N)\Big(\Lambda(\text{softmax}(XW_Q W_K^T X^T))\Big)(XW_K \otimes X) \tag{5}$$

$$\frac{\partial \mathrm{A}(X)}{\partial W_K} = (W_V^T X^T \otimes I_N)\Big(\Lambda(\text{softmax}(XW_Q W_K^T X^T))\Big)(X \otimes XW_Q) \cdot T_{Dd} \tag{6}$$

$$\frac{\partial \mathrm{A}(X)}{\partial W_V} = I_d \otimes \text{softmax}(XW_Q W_K^T X^T)X \tag{7}$$

*where $\Lambda$ is defined in lemma 3.1 and $T_{Dd}$ is the commutation matrix satisfying $T_{Dd}\text{vec}(W_K) = \text{vec}(W_K^T)$ (see section 3.1).*

From proposition 3.1 we obtain a bound on the condition number of the Jacobian of the attention matrix.

**Theorem 3.1.** *Let $\mathrm{A}(X)$ denote a self-attention matrix, as defined in eq.* (1)*, with input $X$ and let $J(\mathrm{A}(X))$ denote its Jacobian with respect to the parameter matrices $W_Q$, $W_K$ and $W_V$ as defined in eq.* (2)*. Assume that $J(\mathrm{A}(X))$ has full rank so that $\kappa(J(\mathrm{A}(X)))$ is finite. Then*

$$\kappa(J(\mathrm{A}(X))) \leq \kappa(X)^3 \big(\Lambda(\text{softmax}(XW_Q W_K^T X^T))\big)\kappa(W_V)\big(\kappa(W_Q) + \kappa(W_K)\big) \tag{8}$$
$$+ \kappa(X)\kappa(\text{softmax}(XW_Q W_K^T X^T))$$

*where $\Lambda$ is defined in lemma 3.1.*

Theorem 3.1 shows that $\kappa(J(\mathrm{A}(X)))$ is bounded above by a sum of two terms:

$$\kappa(X)^3 \cdot \kappa\big(\Lambda(\text{softmax}(XW_Q W_K^T X^T))\big)\kappa(W_V)\big(\kappa(W_Q) + \kappa(W_K)\big) \tag{9}$$

$$\kappa\big(\text{softmax}(XW_Q W_K^T X^T)\big) \tag{10}$$

**Observation.** Theorem 3.1 provides a strategy for reducing the condition number of the Jacobian of A through the upper bound in eq. (8). Since we directly control $W_Q$, $W_K$, and $W_V$, reducing their condition numbers decreases the term in eq. (9), thereby tightening the bound. Our next step is to show that this can be achieved at initialization and to confirm empirically in section 4 that it introduces a bias which improves both optimization and performance.

## 3.3 CONDITIONED INITIALIZATION

Our goal in this section is to design a simple and effective initialization for the query, key, and value matrices $W_Q$, $W_K$, and $W_V$ that lowers the condition number of their singular value spectra, thereby improving the conditioning of the Jacobian J(A).

We begin with two key observations. There are two families of $m \times n$ matrices with condition number 1:

1. scalar multiples of the identity $\lambda I_{m \times n}$ with $\lambda \neq 0$, and
2. semi-orthogonal matrices $\mathcal{O}_{m \times n}$ (matrices with orthonormal rows or columns).

From theorem 3.1, the condition number of J(A) admits the surrogate upper bound

$$\mathcal{B}(\mathrm{J(A)}) := \kappa(X)^3 \, \kappa\big(\Lambda(\text{softmax}(XW_Q W_K^T X^T))\big) \, \kappa(W_V) \, \big(\kappa(W_Q) + \kappa(W_K)\big) \tag{11}$$

$$+ \kappa(X) \, \kappa(\text{softmax}(XW_Q W_K^T X^T)). \tag{12}$$

Unlike the true condition number $\kappa(\mathrm{J(A)})$, the bound $\mathcal{B}(\mathrm{J(A)})$ can be directly influenced at initialization by controlling the conditioning of $W_Q$, $W_K$, and $W_V$. Standard practice initializes these matrices from Gaussian or uniform distributions, which do not enforce good conditioning.

The following proposition shows that initializing $W_Q$, $W_K$, and $W_V$ from either of the families above yields a strictly better surrogate bound at initialization.

> **Proposition 3.2.** *Let* $\mathrm{A}(X)$ *denote an attention matrix with* $W_Q$, $W_K$, *and* $W_V$ *initialized from a Gaussian or uniform distribution, and let* $\overline{\mathrm{A}}(X)$ *denote one where* $\overline{W}_Q$, $\overline{W}_K$, *and* $\overline{W}_V$ *are initialized from either* $\{\lambda I_{D \times d}\}$ *or* $\mathcal{O}_{D \times d}$. *Then*
> $$\mathcal{B}(\mathrm{J}(\overline{\mathrm{A}})) \ \leq \ \mathcal{B}(\mathrm{J}(\mathrm{A})).$$

**Remark.** The quantity $\mathcal{B}(\mathrm{J}(\mathrm{A}))$ is only an upper bound on $\kappa(\mathrm{J}(\mathrm{A}))$. Thus, proposition 3.2 guarantees a tighter bound but does not by itself imply improved conditioning of the Jacobian. Nonetheless, as we demonstrate in section 4, the initialization in proposition 3.2 consistently lowers the condition number during training and leads to more stable optimization and improved performance.

**Initialization Strategy.** Although proposition 3.2 shows that initializing the matrices $W_Q$, $W_K$, and $W_V$ using either $\{\lambda I_{D \times d}\}$ or $\mathcal{O}_{D \times d}$ can potentially lower the condition number of $\mathrm{J}(\mathrm{A})$, it does not specify which choice is most suitable for each matrix. We note that $W_Q$, $W_K$, and $W_V$ play distinct algebraic roles in attention, and this motivates different treatments. For the value map $W_V$, which enters linearly into the output

$$(\text{softmax}(XW_QW_K^TX^T))(XW_V),$$

initializing with the rectangular identity preserves the scale of the input representations ($XW_V = X$), keeps $\kappa(W_V) = 1$, and avoids unnecessary distortion of the Jacobian. In contrast, the query and key maps interact bilinearly through

$$S = XW_QW_K^TX^T,$$

and initializing them as rectangular identities can bias projections toward coordinate subspaces, yielding anisotropic logits and unstable softmax dynamics. A semi-orthogonal initialization for $W_Q$ and $W_K$ instead provides near-isometric embeddings, giving each head balanced representations of $X$ and supporting more diverse and stable attention patterns.

> **Implementation.** Following the above design principle, in the experiments of section 4 we initialized the value matrices $W_V$ for each head as rectangular identities. For the queries and keys, we initialized each $W_Q^{(i)}$ and $W_K^{(i)}$ in the $i$-th head with independent semi-orthogonal projections,
> $$(W_Q^{(i)})^TW_Q^{(i)} = I_d, \qquad (W_K^{(i)})^TW_K^{(i)} = I_d,$$
> for $i = 1, \ldots, h$, where $h$ is the number of heads. This produces near-isometric embeddings into distinct subspaces, thereby diversifying the logits $S^{(i)} = (XW_Q^{(i)})(XW_K^{(i)})^T$ and the resulting attention patterns. See section A.1.1 for a concrete way to carry out this procedure. We refer to this initialization as **conditioned initialization**.

**Different forms of attention.** The formulation in section 3.1 describes the classical self-attention layer used in Transformers. In practice, many recent architectures have proposed variations of attention to improve efficiency and effectiveness (Touvron et al., 2021; Ali et al., 2021; Liu et al., 2021; Ding et al., 2022; Xiong et al., 2021b). In particular, many of these newer forms of attention apply normalization to the query, key and values. Our conditioned initialization is readily applicable to these generalized forms of attention, including those with normalization (see Henry et al. (2020); Dehghani et al. (2023); Zhang & Sennrich (2019)), and we empirically demonstrate in section 4 that it consistently yields strong performance.

## 4 EXPERIMENTS

In this section, we evaluate the theoretical insights from section 3 across a range of Transformer applications. For each setting, we compare our conditioned initialization against the standard default schemes commonly used in the literature, as well as more recent alternatives such as mimetic

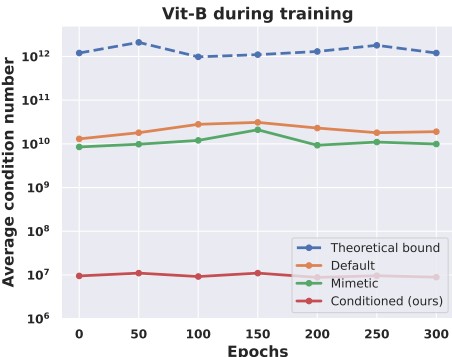 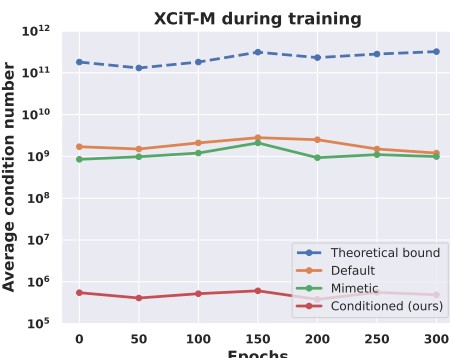

Figure 1: Average condition number of the attention Jacobian during training under three common initialization schemes, shown alongside the theoretical bound from eq. (8).

initialization (Trockman & Kolter, 2023) and weight selection (Xu et al., 2023). In all experiments, conditioned initialization is implemented as described in section 3.3. The goal is to test our initialization on a wide range of architectures at different parameter levels. For larger scale experiments we refer the reader to section A.4.

## 4.1 ViTs for Image Classification

**Vision Transformers.** We applied conditioned initialization to the attention layer of a variety of modern vision Transformers: a ViT-Base (ViT-B) (Dosovitskiy et al., 2020), a Swin-Base (Swin-B) (Liu et al., 2021), a XCiT-Medium (XCiT-M) (Ali et al., 2021), a DeiT-Base (DeiT-B) (Touvron et al., 2021), and a DaViT-Base (DaViT-B) (Ding et al., 2022) for image classification on ImageNet-1k. Each model is initialized in three ways: (i) the default truncated normal initialization (Hugging Face, 2025b), (ii) mimetic initialization (Trockman & Kolter, 2023), and (iii) our conditioned initialization from section 3.3. We point out that each of these vision Transformers uses a different attention layers to the standard self-attention used in the ViT-B architecture. However, as mentioned in section 3.3 conditioned initialization applies to more general forms of attention.

**Results on ImageNet-1k.** We trained three versions of each Transformer: a baseline model with a default initialization (Hugging Face, 2025b), one with mimetic initialization (Trockman & Kolter, 2023) and one incorporating conditioned initialization, see section A.3.1 for training details. The final results, summarized in table 1, show the test accuracy for each initialization on each model. We observe that in every case, conditioned initialization outperforms the other two.

**Validating the theory.** We validate the theoretical results of section 3 using ViT-B and XCiT-M models trained on ImageNet-1k. Figure 1 reports the average condition number of the Jacobian of the attention matrices for ViT-B (left) and XCiT-M (right), during training, under the above mentioned three initializations, alongside the theoretical upper bound from theorem 3.1. The results demonstrate that conditioned initialization consistently yields a better-conditioned Jacobian, providing empirical support for its role in enabling more stable attention mechanisms.

Table 1: Comparison of Vision Transformers with different initializations pretrained on ImageNet-1k. We report Top-1% classification accuracy. In each case, conditioned initialization improves performance over the default and mimetic initializations.

|  | ViT-B | DeiT-B | Swin-B | XCiT-M | DaViT-B |
|---|---|---|---|---|---|
| Original | 80.3 | 81.6 | 83.4 | 82.6 | 84.3 |
| Mimetic | 80.5 | 81.6 | 83.5 | 82.6 | 84.4 |
| Conditioned (ours) | 81.5 | 82.7 | 84.6 | 83.5 | 85.3 |

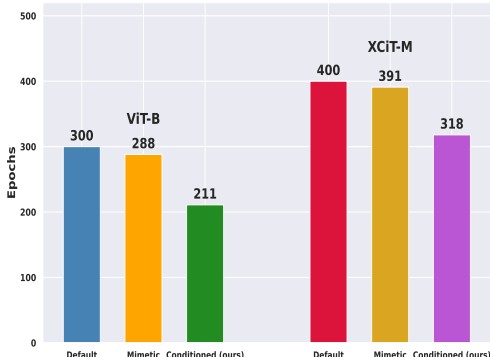 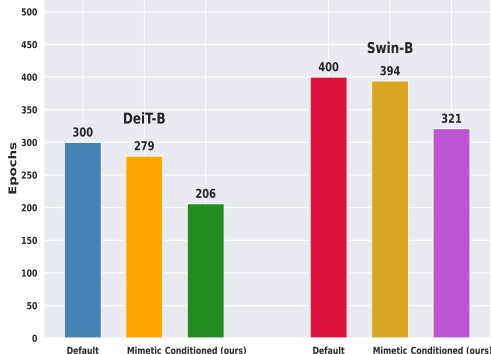

Figure 2: Total number of epochs required for each initialization to reach the final accuracy of the default initialization reported in table 1, across ViT-B, DeiT-B, Swin-B, and XCiT-M. In all cases, conditioned initialization converges more quickly, requiring fewer epochs than both default and mimetic initialization.

**Optimization efficiency.** As shown in table 1, conditioned initialization achieves higher accuracy under the same training regime compared to both default and mimetic initialization. To further assess training efficiency, we measured the number of epochs required by mimetic and conditioned initialization to match the final accuracy attained by the default initialization across ViT-B, DeiT-B, Swin-B, and XCiT-M. Figure 2 reports these results, demonstrating that conditioned initialization consistently converges 20–30% faster to the same accuracy level. Similar analysis for DaViT-B is given in section A.3.1.

### 4.1.1 SMALL SCALE DATASETS

Following section 4.1, we assessed conditioned initialization on small-scale image classification datasets: Flowers (Nilsback & Zisserman, 2008), Pets (Vedaldi, 2012), CIFAR-10, and CIFAR-100 (Krizhevsky et al., 2009). Experiments were conducted with the ViT-Tiny (ViT-T) architecture, a common choice for such benchmarks (Trockman & Kolter, 2023; Xu et al., 2023). We compared conditioned initialization with the default truncated normal scheme (Hugging Face, 2025b) and with mimetic initialization (Trockman & Kolter, 2023). Because ViTs lack strong inductive bias, they typically perform poorly on small datasets; mimetic initialization has been shown to partially remedy this by providing a more suitable inductive bias. As shown in table 2, conditioned initialization reliably improves over the default and performs on par with mimetic initialization.

**Weight selection.** We compared our initialization with the weight selection method from Xu et al. (2023). This can be found in section A.3.1.

Table 2: Comparison of ViT-T pretrained on four small scale datasets with three different initializations. We report Top-1% classification accuracy. In each case, conditioned initialization improves performance over the default and mimetic initializations.

|                     | Pets | Flowers | CIFAR-10 | CIFAR-100 |
|---------------------|------|---------|----------|-----------|
| Default             | 26.7 | 64.5    | 92.4     | 71.7      |
| Mimetic             | 47.7 | 71.6    | 93.6     | 75.0      |
| Conditioned (ours)  | 47.7 | 72.1    | 94.1     | 75.3      |

### 4.2 OBJECT DETECTION AND INSTANCE SEGMENTATION

In this section, we test conditioned initialization in a fine-tuning setting on two downstream tasks: object detection and instance segmentation. We first pretrain an XCiT architecture (Ali et al., 2021) on ImageNet-1K and then fine-tune on COCO 2017 (Lin et al., 2014). The XCiT models are used

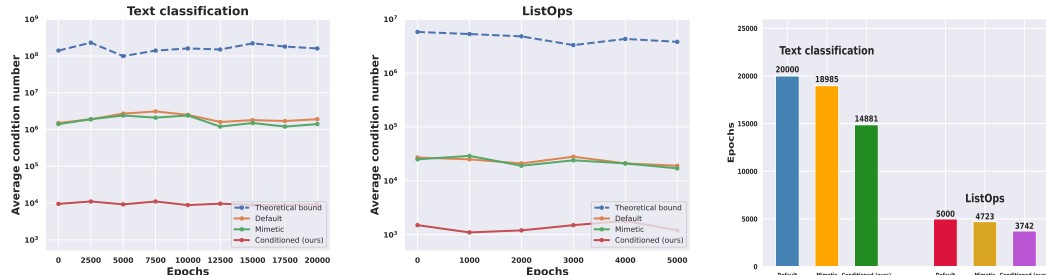

Figure 3: Average condition number of the attention Jacobian during training under three common initialization schemes, shown alongside the theoretical bound from eq. (8).

as backbones within a Mask R-CNN framework (He et al., 2017) equipped with a Feature Pyramid Network (FPN) for multi-scale features. To connect XCiT to FPN, we adapt its column structure to output intermediate representations, using the 12-layer XCiT-Small (XCiT-S) with strides adjusted from 16 to $[4, 8, 16, 32]$ for FPN compatibility. Downsampling is done with max pooling and up-sampling with a single transposed convolution. We evaluate this setup across three initializations: default truncated normal, mimetic, and our conditioned initialization.

**Results.** The results for object detection and instance segmentation are presented in table 3. We report $AP^b$ (Average Precision for bounding boxes), $AP^b_{50/75}$ (Average Precision at IoU thresholds of 0.50 and 0.75 for bounding boxes), $AP^m$ (Average Precision for masks), and $AP^m_{50/75}$ (Average Precision at IoU thresholds of 0.50 and 0.75 for masks). Across all metrics, XCiT models with conditioned initialization consistently outperform both alternatives.

Table 3: Performance evaluation of object detection and instance segmentation on the COCO dataset. For each metric, our spectrally conditioned architecture (Spec. cond.) outperforms the original.

| Model | $AP^b$ | $AP^b_{50}$ | $AP^b_{75}$ | $AP^m$ | $AP^m_{50}$ | $AP^m_{75}$ |
|---|---|---|---|---|---|---|
| Default | 44.9 | 66.1 | 48.9 | 40.1 | 63.1 | 42.8 |
| Mimetic | 44.8 | 66.0 | 49.1 | 40.2 | 63.1 | 42.9 |
| Conditioned (ours) | 45.5 | 66.8 | 49.5 | 40.6 | 63.5 | 43.3 |

### 4.3 LONG RANGE SEQUENCE MODELING

Long-range sequences are essential for Transformers, enabling the integration of information across distant tokens. We assess our initialization scheme on the Long-Range Arena (LRA) benchmark (Tay et al., 2020), designed to evaluate models on extended inputs. For this, we use the Nyströmformer (Xiong et al., 2021b), which achieves efficient long-range modeling via near-linear attention. We train three variants: one with default truncated normal initialization (Xiong et al., 2021b; Hugging Face, 2025a), one with mimetic initialization (Trockman & Kolter, 2023), and one with our conditioned initialization from section 3.3 following the setup of Xiong et al. (2021b).

**Results.** From table 4 we see that across all tasks in the LRA benchmark, conditioned initialization consistently outperforms both default and mimetic initialization. We validate the theoretical results of section 3 on the text classification and ListOps task in the LRA benchmark suite. Figure 3 reports the average condition number of the Jacobian of the attention matrices for a Nyströmformer on the text classification task (left) and a Nyströmformer on the ListOps task (middle), during training, under the above mentioned three initializations, alongside the theoretical upper bound from theorem 3.1. The results demonstrate that conditioned initialization consistently yields a better-conditioned Jacobian, providing empirical support for its role in enabling more stable attention mechanisms. Furthermore, we measured the number of epochs required by mimetic and condi-

tioned initialization to match the final accuracy attained by the default initialization. Figure 3 (right) reports these results demonstrating that conditioned initialization consistently converges approximately 25% faster to the same accuracy level.

Table 4: Nyströmformer with three different initializations on the LRA benchmark. We report evaluation accuracy (%). As shown, our initialization improves performance across all tasks.

| Model | ListOps | Text | Retrieval | Image | Pathfinder |
|---|---|---|---|---|---|
| Default | 37.1 | 63.8 | 79.8 | 39.9 | 72.9 |
| Mimetic | 37.1 | 64.0 | 79.9 | 40.2 | 73.2 |
| Conditioned (ours) | 37.9 | 64.9 | 80.8 | 40.4 | 73.9 |

## 4.4 LANGUAGE MODELING

We apply the insights from section 3 to the Crammed BERT language model (Geiping & Goldstein, 2023), trained with masked language modeling. We consider three variants: the original model with default normal initialization ($\mu = 0, \sigma = 0.02$), one with mimetic initialization, and one with our conditioned initialization. All models are pretrained on The Pile (Gao et al., 2021) following the setup of Geiping & Goldstein (2023), and evaluated on the GLUE benchmark (Wang et al., 2018). As shown in table 5, conditioned initialization yields the strongest performance, with mimetic also outperforming the default baseline. Additional results for GPT-2 are provided in section A.3.4.

Table 5: We evaluate a pretrained Crammed BERT with three different initialization schemes on the GLUE benchmark, and find that conditioned initialization outperforms the other two.

| | MNLI | SST-2 | STSB | RTE | QNLI | QQP | MRPC | CoLA | Avg. |
|---|---|---|---|---|---|---|---|---|---|
| Default | 83.8 | 92.3 | 86.3 | 55.1 | 90.1 | 87.3 | 85.0 | 48.9 | 78.6 |
| Mimetic | 84.1 | 92.5 | 86.5 | 55.1 | 90.3 | 87.5 | 85.1 | 50.1 | 78.9 |
| Conditioned | 84.8 | 92.9 | 86.9 | 55.5 | 91.1 | 87.7 | 86.0 | 51.7 | 79.6 |

## 5 LIMITATIONS

Our conditioned initialization is derived by optimizing an upper bound on the condition number of the self-attention Jacobian, rather than minimizing the Jacobian's condition number directly. The motivation was to examine whether such a bound-based initialization could induce a more favorable optimization bias for training Transformers. While this approach offers useful theoretical guidance and is consistent with the empirical gains we observe, it remains an indirect proxy. Developing methods that can efficiently estimate and control the exact Jacobian conditioning during training would therefore be a valuable direction for future work.

## 6 CONCLUSION

In this paper, we introduced a theoretical framework that relates the conditioning of self-attention Jacobians to the spectral properties of the query, key, and value matrices in Transformer architectures. Building on this insight, we proposed a simple initialization strategy, conditioned initialization, that aims to reduce the condition number of the attention Jacobian at initialization, thereby providing a more favorable inductive bias for optimization. Extensive experiments show that this approach consistently improves performance across a wide range of Transformer models and tasks, including image classification, object detection, language modeling, and long-range sequence learning. [1]

---

[1] Digital tools were used for grammar and formatting only. No large language models contributed to the research, and all findings are original work by the authors.

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

## A  APPENDIX

### ETHICS STATEMENT

This work relies solely on publicly available datasets and does not involve human subjects, personally identifiable information, or sensitive data. The proposed methods are developed exclusively to advance fundamental research in machine learning.

### REPRODUCIBILITY STATEMENT

All experiments in this work were designed with reproducibility in mind. References are provided for any external codebases employed, and full details of training protocols and hardware are described in the appendix. Complete proofs of all theoretical results are also included to allow independent verification.

### USE OF LLMS

This manuscript was prepared with the assistance of LLMs for grammar checking only. No language models were used in conducting the research or drafting the scientific content.

### A.1  THEORETICAL ANALYSIS

In this section, we give the proofs of the lemma and theorems from section 3.2 and the proposition from section 3.3.

**Notation.**  For the convenience of the reader we restate the notation we used in section 3.

Given a matrix $Z \in \mathbb{R}^{m \times n}$ we denote the vectorization of $Z$ by $\text{vec}(Z) \in \mathbb{R}^{mn \times 1}$ (Magnus & Neudecker, 2019). Note that for such a matrix there is a transformation $T_{mn} \in \mathbb{R}^{mn \times mn}$ such that $T_{mn}\text{vec}(Z) = \text{vec}(Z^T)$ where $Z^T$ denotes the transpose of $Z$. The matrix $T_{mn}$ is known as a commutation matrix and is a permutation matrix (Magnus & Neudecker, 2019). The maximum singular value of a matrix $Z$ will be denoted by $\sigma_{\max}(Z)$ and the minimum singular value by $\sigma_{\min}(Z)$. We will use the standard terminology SVD to denote the singular value decomposition of a matrix. Given a vector $v \in \mathbb{R}^n$ the notation $||v||_2$ will denote the vector 2-norm of $v$. Finally, we will let $I_{m \times n}$ denote the rectangular identity matrix that has all 1's on its main diagonal. When we are dealing with square matrices we will often simply write $I_n$ with the understanding that $I_n$ is the $n \times n$ square identity matrix. Context will make it clear whether we are in the square or non-square regime. We also let $\mathcal{O}_{m \times n}$ as the real $m \times n$ semi-orthogonal matrices.

We start with the following standard facts on derivatives of matrices. We point out to the reader that Qi et al. (2025) also computes various Jacobians of the self-attention layer using Kronecker factorizations however their computations are slightly different to ours and we therefore give explicit details of how to obtain the proofs of lemma 3.1 and proposition 3.1.

**Lemma A.1.** *Let $A \in R^{n \times m}$, $B \in \mathbb{R}^{k \times l}$ and $C \in \mathbb{R}^{m \times k}$. Then*

$$\frac{\partial ACB}{\partial C} = B^T \otimes A. \tag{13}$$

*Proof.* We start by using a well known vectorization identity (Magnus & Neudecker, 2019)

$$\text{vec}(ACB) = (B^T \otimes A)\text{vec}(C) \tag{14}$$

where $\text{vec}$ denotes the vectorization operator which takes a matrix and maps it to a vector by stacking its columns on top of each other, see Magnus & Neudecker (2019). We then differentiate eq. (14) to obtain

$$\frac{\partial \text{vec}(ACB)}{\partial \text{vec}(C)} = B^T \otimes A. \tag{15}$$

The result of the lemma follows.  □

**Lemma A.2.** *Let $A \in \mathbb{R}^{n \times m}$ so that $A^T \in \mathbb{R}^{m \times n}$. Then*

$$\text{vec}(A^T) = T_{mn}\text{vec}(A) \tag{16}$$

$$\frac{\partial \text{vec}(A^T)}{\partial \text{vec}(A)} = T_{mn} \tag{17}$$

*where $T_{mn}$ is a commutation matrix.*

*Proof.* The first equation follows from the definition of the transpose of a matrix (Magnus & Neudecker, 2019). The second equation then follows from the first. $\square$

The proof of lemma 3.1 is given as follows.

*Proof of lemma 3.1.* Let $[z_1, \ldots, z_n]$ denote a row vector in $\mathbb{R}^n$. Then by definition

$$\text{softmax}([z_1, \ldots, z_n]) = \left[ \frac{e^{z_1}}{\sum_{i=1}^n e^{z_i}}, \ldots, \frac{e^{z_n}}{\sum_{i=1}^n e^{z_i}} \right]. \tag{18}$$

From the above equation we can compute the partial derivative and find

$$\frac{\partial \text{softmax}(z)_i}{\partial z_j} = \text{softmax}(z)_i \left( \delta_{ij} - \text{softmax}(z)_j \right). \tag{19}$$

The term $\frac{\partial \text{softmax}(z)_i}{\partial z_j}$ is precisely the $ij$ component of the matrix $\frac{\partial \text{softmax}(z)}{\partial z}$. Putting each of these $ij$ terms into an $n \times n$ matrix we find

$$\frac{\partial \text{softmax}(z)}{\partial z} = Diag(z) - z \cdot z^T \tag{20}$$

which proves the lemma. $\square$

We can use the above lemmas to give the proof of proposition 3.1.

*Proof of proposition 3.1.* We start by establishing the derivative formula for the term $\frac{\partial A(X)}{\partial W_Q}$. We will use the notation used in section 3.1. Note that by definition $\frac{\partial A(X)}{\partial W_Q} \in \mathbb{R}^{dN \times dD}$. Using eq. (1)

$$A(X) = I_N \text{softmax}(XW_Q W_K^T X^T) XW_V \tag{21}$$

where $I_N$ is the $N \times N$ identity matrix. This is done so that we can apply lemma A.1. We then compute

$$\frac{\partial A(X)}{\partial W_Q} = \frac{\partial (I_N \text{softmax}(XW_Q W_K^T X^T) XW_V)}{\partial W_Q} \tag{22}$$

$$= (W_V^T X^T \otimes I_N) \frac{\partial \text{softmax}(XW_Q W_K^T X^T)}{\partial W_Q} \text{ using } lemma \ A.1 \tag{23}$$

$$= (W_V^T X^T \otimes I_N) \Lambda(\text{softmax}(XW_Q W_K^T X^T)) \frac{\partial (XW_Q W_K^T X^T)}{\partial W_Q} \text{ using } lemma \ 3.1 \tag{24}$$

$$= (W_V^T X^T \otimes I_N) \Lambda(\text{softmax}(XW_Q W_K^T X^T))(XW_K \otimes X) \tag{25}$$

which proves the firs equality in proposition 3.1.

To compute $\frac{\partial A(X)}{\partial W_K} \in \mathbb{R}^{dN \times dD}$ we proceed in a similar way.

$$\frac{\partial A(X)}{\partial W_K} = \frac{\partial (I_N \text{softmax}(XW_Q W_K^T X^T) XW_V)}{\partial W_K} \tag{26}$$

$$= (W_V^T X^T \otimes I_N) \frac{\partial \text{softmax}(XW_Q W_K^T X^T)}{\partial W_K} \text{ using } lemma \ A.1 \tag{27}$$

$$= (W_V^T X^T \otimes I_N) \Lambda(\text{softmax}(XW_Q W_K^T X^T)) \frac{\partial (XW_Q W_K^T X^T)}{\partial W_K} \text{ using } lemma \ 3.1 \tag{28}$$

$$= (W_V^T X^T \otimes I_N) \Lambda(\text{softmax}(XW_Q W_K^T X^T))(X \otimes XW_Q) T_{Dd} \tag{29}$$

where the last equality follows from lemmas A.1 and A.2 This establishes the second equality in proposition 3.1.

To prove the identity for $\frac{\partial A(X)}{\partial W_V} \in \mathbb{R}^{dN \times dD}$ we write

$$A(X) = \text{softmax}(XW_Q W_K^T X^T)XW_V I_d \tag{30}$$

where $I_d$ is the $d \times d$ identity matrix. Then we simply apply lemma A.1 to obtain

$$\frac{\partial A(X)}{\partial W_V} = \frac{\partial(\text{softmax}(XW_Q W_K^T X^T)XW_V I_d)}{\partial W_V} \tag{31}$$

$$= I_d \otimes \text{softmax}(XW_Q W_K^T X^T)X \tag{32}$$

which proves the final equality in proposition 3.1. $\qquad\square$

We also give the proof of theorem 3.1.

*Proof of theorem 3.1.* The proof of theorem 3.1 follows from using proposition 3.1 and the definition of the Jacobian of $A(X)$ with respect to $W_Q$, $W_K$ and $W_V$ given by

$$J(A(X)) = \left[ \frac{\partial(A(X))}{\partial W_Q}, \frac{\partial(A(X))}{\partial W_K}, \frac{\partial(A(X))}{\partial W_V} \right]^T. \tag{33}$$

We recall that the condition number is defined as $\kappa(J(A(X))) = \frac{\sigma_{\max}(J(A(X)))}{\sigma_{\min}(J(A(X)))}$ where $\sigma_{\max}(J(A(X)))$ is the maximum singular value of $J(A(X))$ and $\sigma_{\min}(J(A(X)))$ the minimum singular value which we know is non-zero because of the assumption that $J(A(X))$ has full rank. Note that, using the notation in section 3.1, we have that $J(A(X)) \in \mathbb{R}^{3dN \times dD}$ as $\frac{\partial A(X)}{\partial W_Q}, \frac{\partial A(X)}{\partial W_K}, \frac{\partial A(X)}{\partial W_V} \in \mathbb{R}^{dN \times dD}$. For each of notation we will write $A_Q := \frac{\partial A(X)}{\partial W_Q}, A_K := \frac{\partial A(X)}{\partial W_K}, A_V := \frac{\partial A(X)}{\partial W_V}$.

We will start by computing a bound for the maximum singular value. We have for any vector $z \in \mathbb{R}^{dD}$ we have

$$||J(A(X))z||_2^2 = \left\| \left[ \frac{\partial(A(X))}{\partial W_Q}(z), \frac{\partial(A(X))}{\partial W_K}(z), \frac{\partial(A(X))}{\partial W_V}(z) \right]^T \right\|_2^2 \tag{34}$$

$$= \left\| \left[ A_Q(z), A_K(z), A_V(z) \right] \right\|_2^2 \tag{35}$$

$$= ||A_Q(z)||_2^2 + ||A_K(z)||_2^2 + ||A_V(z)||_2^2 \tag{36}$$

$$\leq \left( \sigma_{\max}(A_Q)^2 + \sigma_{\max}(A_K)^2 + \sigma_{\max}(A_V)^2 \right) ||z||_2^2. \tag{37}$$

This implies that

$$\sigma_{\max}(J(A(X))) := \max_{z \neq 0} \frac{||J(A(X))z||_2^2}{||z||_2^2} \tag{38}$$

$$\leq \sqrt{\sigma_{\max}(A_Q)^2 + \sigma_{\max}(A_K)^2 + \sigma_{\max}(A_V)^2} \tag{39}$$

$$\leq \sigma_{\max}(A_Q) + \sigma_{\max}(A_K) + \sigma_{\max}(A_V). \tag{40}$$

The next step is to compute a lower bound for the minimum singular value $\sigma_{\min}(J(A(X)))$. The approach is similar to the above, using the fact that $\sigma_{\min}(J(A(X))) = \min_{||z||_2=1} J(A(X)(z)$. We can then use the inequality

$$||J(A(X)(z)||_2^2 = \left\| \left[ \frac{\partial(A(X))}{\partial W_Q}(z), \frac{\partial(A(X))}{\partial W_K}(z), \frac{\partial(A(X))}{\partial W_V}(z) \right]^T \right\|_2^2 \tag{41}$$

$$\geq \max \left\{ \left\| \frac{\partial(A(X))}{\partial W_Q}(z) \right\|, \left\| \frac{\partial(A(X))}{\partial W_K}(z) \right\|, \left\| \frac{\partial(A(X))}{\partial W_V}(z) \right\| \right\} \tag{42}$$

Then minimizing the above over the constraint $z \in \mathbb{R}^{dD}$ such that $||z|| = 1$ we obtain

$$\sigma_{\min}(J(A(X))) \geq \max \left\{ \sigma_{\min}\left( \frac{\partial(A(X))}{\partial W_Q} \right), \sigma_{\min}\left( \frac{\partial(A(X))}{\partial W_K} \right), \sigma_{\min}\left( \frac{\partial(A(X))}{\partial W_V} \right) \right\}. \tag{43}$$

Combing the bounds on $\sigma_{\max}(J(A(X)))$ and $\sigma_{\min}(J(A(X)))$ we obtain

$$\kappa(J(A(X))) \leq \kappa\left(\frac{\partial(A(X))}{\partial W_Q}\right) + \kappa\left(\frac{\partial(A(X))}{\partial W_K}\right) + \kappa\left(\frac{\partial(A(X))}{\partial W_V}\right). \tag{44}$$

The final step is to get a bound on the condition numbers for each term on the right hand side in the above inequality. This is done using two facts: Firstly, given two matrices $C$ and $D$ such that the product $CD$ is full rank then $\kappa(CD) \leq \kappa(C)\kappa(D)$ and the second that $\kappa(C \otimes D) = \kappa(C)\kappa(D)$. Using these two facts we can then use proposition 3.1 to obtain the bound of theorem 3.1 and the proof is finished. $\square$

Using theorem 3.1 the proof of proposition is straightforward.

*Proof of proposition 3.2.* We first observe that by definition of the condition number of a general $m \times n$ matrix $M$ must always satisfy $\kappa(M) \geq 1$. For matrices with 1's on the diagonal and zero elsewhere the condition number is 1. If $M$ is a semi-orthogonal matrix then either

$$1.\ MM^T = I_{m \times m} \text{ if } m \geq n \tag{45}$$

$$2.\ M^T M = I_{n \times n} \text{ if } n \geq m. \tag{46}$$

Since the singular values of $M$ are precisely the eigenvalues of $MM^T$ if $m \geq n$ or $M^T M$ if $n \geq m$. It follows that the singular values of $M$ must all be 1 and hence it has condition number 1.

Therefore, if the $W_Q$, $W_K$ and $W_V$ matrices are initialized as $I_{D \times d}$ or as a matrix in $\mathcal{O}_{D \times d}$ we have that their condition number is 1. Therefore, we must have

$$\mathcal{B}(\overline{A}) \leq \mathcal{B}(A) \tag{47}$$

and the proposition is proved. $\square$

*Remark* A.1. In the implementation strategy in section 3.2 we saw that we initialized the values matrix $W_V$ with a rectangular identity matrix $I_{D \times d}$. However, from the theory of that section we could have also initialized it with $\lambda I_{D \times d}$ for $\lambda \neq 0$. In general, we found empirically that this could be done but that if $\lambda$ got too large we noticed some instability in training due to $W_V$ having weights that were too large. Therefore, opting for the identity $I_{D \times d}$ was what we found worked well.

**Why Conditioning the Jacobian Aids Optimization.** The rationale for improving the conditioning of the self-attention Jacobian is connected to established results on the Neural Tangent Kernel (NTK). Prior work, see Liu et al. (2022), has shown that better-conditioned NTKs lead to faster and more reliable convergence to a global minima during gradient-based optimization. Since the singular values of a network's Jacobian correspond to the positive square roots of the eigenvalues of its NTK, improving the conditioning of the Jacobian directly enhances the conditioning of the NTK. This connection provides a theoretical basis for why controlling the spectral structure of the self-attention Jacobian can benefit optimization. Furthermore, recent extensions of NTK theory to transformer architectures (e.g., Yang 2020) support the relevance of these insights in the attention setting. Our initialization scheme leverages this relationship by explicitly targeting improved conditioning at initialization, which is consistent with the empirical performance gains observed across tasks.

### A.1.1 IMPLEMENTATION DETAILS

In this section, we give the details of how we implement the semi-orthogonal initialization of the $W_Q$ and $W_V$ matrix of each head.

We recall from section 3.3 that our initialization for the queries and keys, proceeded by initializing each $W_Q^{(i)}$ and $W_K^{(i)}$ in the $i$-th head with independent semi-orthogonal projections,

$$(W_Q^{(i)})^T W_Q^{(i)} = I_d, \qquad (W_K^{(i)})^T W_K^{(i)} = I_d,$$

for $i = 1, \ldots, h$, where $h$ is the number of heads. To do this suppose each $W_Q^{(i)}$ and $W_K^{(i)}$ are $D \times d$ and let $r = \min(D, d)$ for each head we form two random matrices $R_i^Q$ and $R_i^K$ of shape $D \times d$ and then take the truncated SVD

$$U_i^Q(r)S_i^Q(r)(V_i^Q)^T(r) \text{ and } U_i^K(r)S_i^K(r)(V_i^K)^T(r) \tag{48}$$

where each of the $U_i^Q(r) \in \mathbb{R}^{D \times r}$ and $(V_i^Q)(r) \in \mathbb{R}^{r \times d}$ and similarly for the $K$ ones. We then observe that

$$O_i^Q := U_i^Q(r) \cdot (V_i^Q)^T(r) \text{ and } O_i^K := U_i^K(r) \cdot (V_i^K)^T(r) \tag{49}$$

are semi-orthogonal. Doing this for each different head we get different semi-orthogonal matrices for each $W_Q$ and $W_K$ for each head. The implementation of $W_V$ is the same for each head and is simply done by fixing $W_V = I_{D \times d}$.

*Remark* A.2. We note that in the above we used the SVD to obtain the initializations of each $W_Q$ and $W_V$. One can also use the QR decomposition (Magnus & Neudecker, 2019) and PyTorch has a built in way to do this via `nn.init.orthogonal`.

## A.2 DISCUSSION ON NORMALIZATION

This section provides additional clarification on how normalization and stabilization mechanisms interact with the Jacobian conditioning analysis and with the proposed conditioned initialization scheme. Several modern transformer architectures apply normalization directly to the queries, keys, or values (e.g., RMSNorm, QKNorm), while Layer Normalization (LN) is typically applied to the inputs of each block. The discussion below outlines how these components relate to the theoretical results in section 3 and to the practical behaviour observed in section 4.

**Theoretical setting.** In the original self-attention formulation Vaswani et al. (2017), no normalization is applied directly to the query, key, or value weight matrices. The only modification is the fixed scaling factor $1/\sqrt{d}$, determined by the head dimension. Since this factor does not depend on the model parameters, differentiating the attention map simply scales the Jacobian by a constant. A constant scalar multiplication does not change the condition number of a matrix; therefore, this scaling has no effect on the conditioning analysis developed in Section 3. The derivations for vanilla self-attention thus remain mathematically correct and serve as a foundation for the initialization scheme proposed later.

**Compatibility with modern normalization layers.** Many contemporary transformer variants incorporate normalization directly into the attention pathway, for example through RMSNorm or QKNorm applied to queries, keys, or values. The conditioned initialization introduced in this paper is designed to operate on top of these mechanisms rather than as a replacement for them. In all experiments, the architectures retain their original normalization layers exactly as implemented in the publicly released codebases. For example, the DeiT-B model applies QKNorm to both queries and keys. When applying conditioned initialization, these QKNorm layers remain in place. This ensures that comparisons in section 4 are consistent with the established baseline implementations in the literature.

**Normalization versus conditioning.** Normalization and conditioning address different aspects of stability. Normalization controls the scale of activations and gradients, whereas conditioning concerns the spectral structure of the Jacobian. These effects are distinct. A matrix may have small norm but poor conditioning, or it may have large norm yet be well-conditioned. For example,

$$A = \begin{bmatrix} 1 & 0 \\ 0 & 1 \end{bmatrix}, \qquad B = \begin{bmatrix} 10.1 & 0 \\ 0 & 10 \end{bmatrix}$$

have very different Frobenius norms but nearly identical condition numbers, while

$$C = \begin{bmatrix} 1 & 0 \\ 0 & 1 \end{bmatrix}, \qquad D = \begin{bmatrix} \sqrt{2} & 0 \\ 0 & 0.1 \end{bmatrix}$$

have similar Frobenius norms but condition numbers 1 and approximately 14.14, respectively. Normalization schemes such as LN, RMSNorm, and QKNorm regulate magnitudes, while conditioned initialization directly shapes the spectral behaviour of the Jacobian. These mechanisms therefore complement one another.

**Ablation on Partial Initialization.** We conducted several ablation studies in which the conditioned initialization was applied to only a subset of the attention projection matrices, such as initializing only $W_Q$, only $W_K$, or only $W_V$. An initial hypothesis was that initializing $W_V$ alone

might be sufficient, since the softmax operation effectively performs a row-wise normalization on the attention scores. However, normalization and conditioning target fundamentally different properties: a matrix may have small norm yet still exhibit a large condition number. This observation led us to a more complete analysis, formalized in theorem 3.1, which shows that the conditioning of the self-attention Jacobian depends on the joint spectral structure of $W_Q$, $W_K$, and $W_V$. Consistent with this theoretical insight, we found that applying conditioned initialization to all three projections yields the most stable behaviour and the strongest empirical performance.

**Discussion on the Output Projection.** We also examined whether the conditioned initialization should be applied to the output projection matrix $W_O$, given its close relationship to $W_V$. Empirically, we found that once $W_V$ is initialized using the proposed scheme, the initialization of $W_O$ has negligible effect. Applying the conditioned initialization to $W_O$ in addition to $W_V$ did not alter training behaviour or final performance. Consequently, our method focuses on conditioning the query, key, and value projections, while leaving $W_O$ with its standard initialization.

## A.3 EXPERIMENTS

### A.3.1 VISION TRANSFORMERS

**Hardware and implementation.** The image classification experiments in section 4.1 of the paper were done on Nvidia A100 GPUs. The implementation of the ViTs was all done using the Timm code base (Wightman, 2019). The architectures were all trained from scratch on the ImageNet-1k dataset using the AdamW optimizer following the hyperparameters used in the original papers (Dosovitskiy et al., 2020; Steiner et al., 2016; Liu et al., 2021; Ali et al., 2021; Touvron et al., 2021; Ding et al., 2022). For the case of ViT-T on the Pets, Flowers, CIFAR-10 and CIFAR-100 datasets we used Wightman (2019) for the architecture and the training hyperparameters from Xu et al. (2023).

**Optimization analysis for DaViT-B.** As shown in table 1, conditioned initialization achieves higher accuracy under the same training regime compared to both default and mimetic initialization on the DaViT-B architecture. To further assess training efficiency, we measured the number of epochs required by mimetic and conditioned initialization to match the final accuracy attained by the default initialization across on the DaViT-B architecture. Figure 4 reports these results, demonstrating that conditioned initialization consistently converges to approximately 25% faster to the same accuracy level.

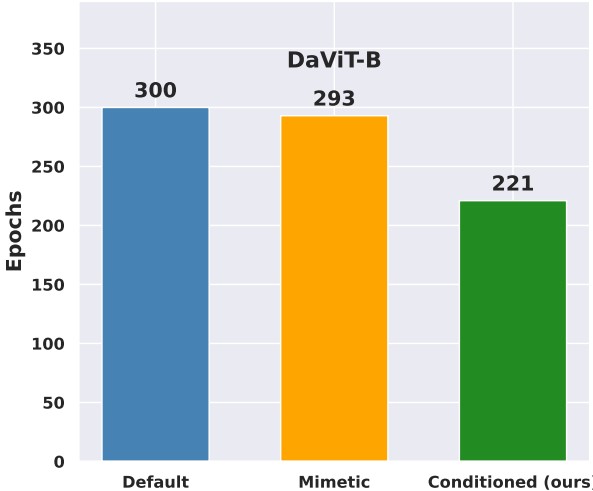

Figure 4: Total number of epochs required for each initialization to reach the final accuracy of the default initialization reported in table 1 for DaViT-B architecture. Conditioned initialization converges more quickly, requiring fewer epochs than both default and mimetic initialization.

**Weight Selection**   Xu et al. (Xu et al., 2023) introduced the weight selection method, showing that weights transferred from an ImageNet-21K-pretrained ViT-Small (ViT-S) model provide a strong inductive bias for initializing a ViT-Tiny (ViT-T) on small-scale datasets. Here, we examine whether weight selection remains effective when pretraining is performed on ImageNet-1K. To this end, we pretrained ViT-S models on ImageNet-1K under three initialization schemes: default truncated normal (Hugging Face, 2025b), mimetic (Trockman & Kolter, 2023), and our conditioned initialization from section 3.3. Using the weight selection procedure (Xu et al., 2023), we then derived corresponding ViT-T initializations and trained them on Food-101 (Bossard et al., 2014), CIFAR-10, and CIFAR-100 (Krizhevsky et al., 2009). Table 6 summarizes the results. Each entry labeled "ImageNet-1K" indicates that the ViT-S model was pretrained on ImageNet-1K with the specified initialization before its weights were transferred to ViT-T via weight selection. For comparison, the final row reports the performance of a ViT-T initialized from an ImageNet-21K-pretrained ViT-S with default truncated normal initialization. The results highlight a key finding: replacing ImageNet-21K with ImageNet-1K can yield comparable performance, provided that the initialization is chosen carefully. In particular, conditioned initialization on ImageNet-1K achieves accuracy on par with default initialization pretrained on ImageNet-21K, underscoring its effectiveness in data-limited pretraining regimes.

Table 6: Comparison of ViT-T pretrained on four small scale datasets with three different initializations. We report Top-1% classification accuracy. In each case, conditioned initialization improves performance over the default and mimetic initializations.

|  | Food-101 | CIFAR-10 | CIFAR-100 |
|---|---|---|---|
| ImageNet-1k + Default | 85.5 | 96.6 | 79.7 |
| ImageNet-1k + Mimetic | 86.4 | 96.3 | 79.9 |
| ImageNet-1k + Conditioned (ours) | **87.3** | **97.1** | 81.0 |
| ImageNet-21k + Default | 87.1 | **97.1** | **81.1** |

**Training loss curves.**   We plot the training curves for each of the vision transformer experiments with each different initialization. In fig. 5, fig. 6, fig. 7 we plot the training loss for the different initializations for the ViT-B, DeiT-B, Swin-B, XCiT-M and DaViT-B architectures respectively. In each case, we see conditioned initialization has trains stably and converges faster when compared to the default and Mimetic initializations.

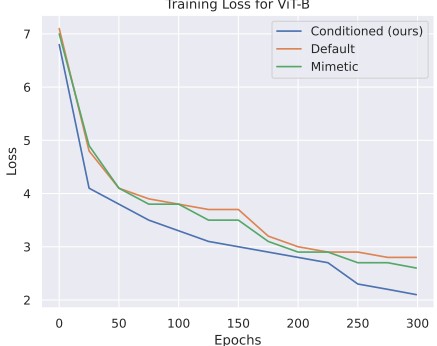 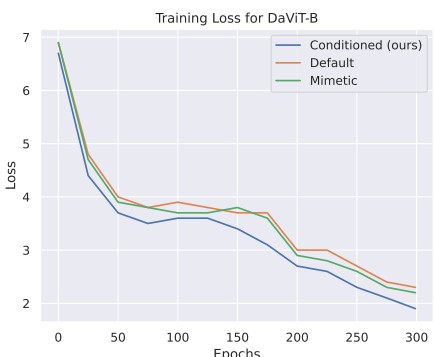

Figure 5: Training loss curves for different initializations for ViT-B (left) and DeiT-B (right).

### A.3.2   OBJECT DETECTION AND INSTANCE SEGMENTATION

**Hardware and Implementation:**   The experiments for section 4.2 of the paper on object detection and instance segmentation were carried out on Nvidia A100 GPUs. The implementation followed He et al. (2017). We used the code base given by the GitHub Matterport (2017) following their exact training regime.

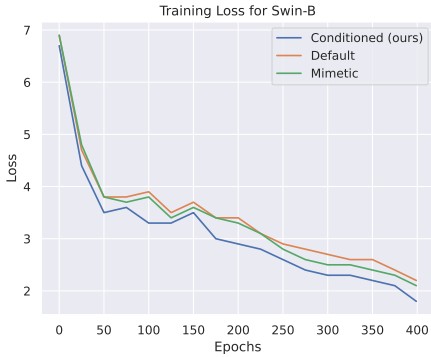 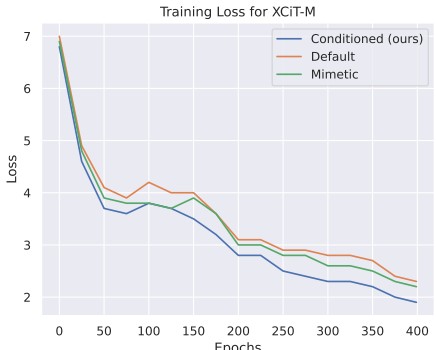

Figure 6: Training loss curves for different initializations for Swin-B (left) and XCiT-M (right).

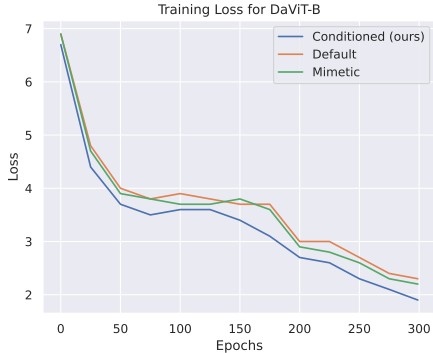

Figure 7: Training loss curves for different initializations for DaViT-B.

### A.3.3 LONG RANGE SEQUENCE MODELING WITH NYSTRÖMFORMER

**Hardware and Implementation.**    All the experiments for the Nyströmformer on LRA benchmark results in section 4.3 were carried out on Nvidia A100 GPUs following the implementation and hyperparameter settings given in Xiong et al. (2021a).

### A.3.4 LANGUAGE MODELING

**Crammed BERT.**    The BERT language modeling experiment in section 4.4 were all carried out on a Nvidia A6000 GPU. The Crammed-Bert was implemented following the original paper Geiping & Goldstein (2023) and the original GitHub Geiping (2023). The training regime follows Geiping (2023).

**GPT-2 on TinyStories.**    We train an autoregressive Transformer, namely a GPT-2 architecture trained on the TinyStories dataset (Eldan & Li, 2023). Once again we compared three initializations: one with the default normal initialization ($\mu = 0, \sigma = 0.02$), one with mimetic and a third with conditioned initialization. As shown in table 7, conditioned initialization achieves a lower perplexity than both the default and mimetic initializations showing that it boosts performance in the setting of autoregressive Transformers. We used the training regime from (Eldan & Li, 2023).

### A.4 LARGER SCALE EXPERIMENTS

In this section we tested our initialization on much larger models in both the vision and language setting.

Table 7: GPT-2 models trained on the TinyStories dataset. We initialize a model in three different ways. Conditioned initialization achieves a perplexity than the other two initializations.

|  | Perplexity |
|---|---|
| Default | 2.47 |
| Mimetic | 2.40 |
| Conditioned (ours) | 2.29 |

We ran a larger GPT-2 model, 472 million parameters, on the WikiText-103 dataset and the TinyStories dataset. The results are shown in table 8 and table 9 below clearly shows our initialization obtains better performance by obtaining the lowest perplexity.

Table 8: Perplexity of 472M GPT-2 model pretrained on TinyStories.

| Model | Perplexity |
|---|---|
| Default | 2.39 |
| Mimetic | 2.32 |
| Conditioned (ours) | 2.20 |

Table 9: Perplexity of 472M GPT-2 model pretrained on WikiText-103.

| Model | Perplexity |
|---|---|
| Default | 44.2 |
| Mimetic | 43.8 |
| Conditioned (ours) | 42.7 |

We repeated this experiment with a 1.72 billion parameter GPT-2 model. In this case we witnessed overfitting in all initialized cases. However, this is to be expected as a billion parameter model is too large for the TinyStories and WikiText-103 datasets. Yet even in this case our initialization performed much better as can be seen from table 10 and table 11.

Table 10: Perplexity of 1.72B GPT-2 model pretrained on TinyStories.

| Model | Perplexity |
|---|---|
| Default | 4.15 |
| Mimetic | 4.25 |
| Conditioned (ours) | 4.03 |

Table 11: Perplexity of 1.72B GPT-2 model pretrained on WikiText-103.

| Model | Perplexity |
|---|---|
| Default | 48.1 |
| Mimetic | 48.5 |
| Conditioned (ours) | 46.9 |

We also ran experiments on large scale ViTs on the ImageNet-1k dataset, where these models range from 200-300M parameters. Once again for these larger models we witnessed overfitting (this has also been observed in Dosovitskiy et al. (2020); Ding et al. (2022)) yet even in this case our initialization outperformed the other two standard ones, see table 12.

We then ran two 1 billion parameter models namely a ViT-Giant (ViT-G) and a DaViT-Giant (DaViT-G) as these where some of the standard billion parameter vision transformers we could find in

Table 12: Large scale ViTs with different initializations pretrained on ImageNet-1k. We show the Top1% accuracy.

|  | ViT-L | DeiT-L | Swin-L | XCiT-L | DaViT-L |
|---|---|---|---|---|---|
| Original | 79.6 | 80.7 | 82.6 | 81.5 | 83.3 |
| Mimetic 2 | 79.7 | 80.6 | 82.4 | 81.5 | 83.2 |
| Conditioned (ours) | 80.7 | 81.5 | 83.7 | 82.4 | 84.4 |

Hugging Face (2025b). We pretrained them each with the different initializations on ImageNet-1k. As can be seen from table 13 our initialization out performs the other two.

Table 13: 1B scale ViTs with different initializations pretrained on ImageNet-1k. We show the Top1% accuracy.

|  | ViT-G | DaViT-G |
|---|---|---|
| Original | 78.8 | 81.9 |
| Mimetic 2 | 78.6 | 82.0 |
| Conditioned (ours) | 79.9 | 82.9 |

