# OpenReview forum: "Conditioned Initialization for Attention"
_ICLR.cc/2026/Conference — ICLR 2026 Poster_

### Official Review · Reviewer_jfhc · 2025-10-22

**Soundness:** 3
**Presentation:** 4
**Contribution:** 3
**Rating:** 6
**Confidence:** 3

**Summary:**

This paper focuses on the initialization of transformers, specifically the initialization of the attention mechanism. Specifically, arguing that for better optimization of the network, the initialization should bound the condition number of the Jacobian matrix of the attention computation.

The paper suggests a practical implementation for this by initializing the $\mathbf{W}_q$ and $\mathbf{W}_k$ matrices as orthogonal matrices, and $\mathbf{W}_v$ matrix as an identity one.

Through a series of experiments across language and vision tasks, the initialization is shown to achieve better performance, while achieving the same accuracy as baselines with significantly less compute, highlighting its efficiency.

**Strengths:**

S1: The paper is clear and can be followed easily.

S2: The practical implementation of the initialization is easy to use, making it conducive to wide adoption.

S3: The method shows consistent performance gains compared to the baselines across different tasks and modalities (language and vision).

S4: The suggested initialization method is highly efficient, achieving the same performance as baselines with much less compute. This is an advantage that could be highlighted further by the authors, perhaps through showing detailed training loss curves of the different models and adding more figures like Figure 2 for all tasks.

**Weaknesses:**

W1: The main weakness is that while the paper demonstrates the proposed initialization bounds the condition number of the Jacobian matrix of the attention computation, it does not sufficiently explain the theoretical basis for why this is a desired property for better optimization. Although better performance is shown in practice, a proof or a more robust theoretical explanation for this optimization benefit is expected, especially since the paper outlines how to achieve the bound.

W2: This is a minor point, but using the notation $\mathbf{A}(\mathbf{X})$ for the output of the attention computation is potentially confusing, as this symbol is often reserved for the attention matrix itself. A symbol change is suggested for clarity.

**Questions:**

See W1

---

> ### Author Response · Authors · 2025-11-21
> **Response to  Reviewer jfhc**
>
> We thank the reviewer for the time they have taken to review our paper. We answer all the points and questions they have raised in their review below. We have also updated our paper and added further sections and points in the paper that address the points/questions raised by the reviewer.
>
> **Reviewer jfhc: The main weakness is that while the paper demonstrates the proposed initialization bounds the condition number of the Jacobian matrix of the attention computation, it does not sufficiently explain the theoretical basis for why this is a desired property for better optimization. Although better performance is shown in practice, a proof or a more robust theoretical explanation for this optimization benefit is expected, especially since the paper outlines how to achieve the bound.**
>
>  We thank the reviewer for their comment. This is partly explained in the Related Work section, Section 2, and is based on the Neural Tangent Kernel. The rationale for improving the conditioning of the self-attention Jacobian is connected to established results on the Neural Tangent Kernel (NTK). Prior work, see Liu et al., has shown that better-conditioned NTKs lead to faster and more reliable convergence to a global minima during gradient-based optimization. Since the singular values of a network's Jacobian correspond to the positive square roots of the eigenvalues of its NTK, improving the conditioning of the Jacobian directly enhances the conditioning of the NTK. This connection provides a theoretical basis for why controlling the spectral structure of the self-attention Jacobian can benefit optimization. Furthermore, recent extensions of NTK theory to transformer architectures (e.g., Yang 2020) support the relevance of these insights in the attention setting. Our initialization scheme leverages this relationship by explicitly targeting improved conditioning at initialization, which is consistent with the empirical performance gains observed across tasks. We have added a new part to the appendix that discusses this for the reader with the relevant citations, see lines 840-850.
>
> **Reviewer jfhc: The suggested initialization method is highly efficient, achieving the same performance as baselines with much less compute. This is an advantage that could be highlighted further by the authors, perhaps through showing detailed training loss curves of the different models and adding more figures like Figure 2 for all tasks.**
>
> Thank you for this comment. We have added training curves for the main ViT experiments trained on ImageNet-1k in the appendix in section A.3.1. In each case we see that our initialization converges faster than the other ones.
>
> **Reviewer jfhc: This is a minor point, but using the notation $A(X)$
>  for the output of the attention computation is potentially confusing, as this symbol is often reserved for the attention matrix itself. A symbol change is suggested for clarity.**
>
> We thank the reviewer for pointing this out and we apologize for this confusion. The reason we used that notation is that often people speak of the attention layer in a transformer and this layer is defined by using all the query, key and value parts of the layer. Since our initialization works on all the components we used that notation to define the whole layer. We do understand this can cause confusion so we have added a statement in the Preliminaries section, section 3.1 of the paper, clearly stating this so as to avoid confusion, see lines 120-122.

---

> > ### Comment · Reviewer_jfhc · 2025-11-25
> >
> > Thank you for your detailed response. I think the additional part added regarding the theoretical basisis important for the paper.
> >
> > I will maintain my (positive) score.

---

### Official Review · Reviewer_44Pd · 2025-10-29

**Soundness:** 4
**Presentation:** 3
**Contribution:** 4
**Rating:** 8
**Confidence:** 5

**Summary:**

This paper proposes to initialize the query and key weights of self-attention layers as semi-orthogonal matrices, and the value weights as rectangular identity matrices. This scheme is based on an analysis of the condition number of the Jacobian of self-attention layers, supported by previous research suggesting that bounding this condition number leads to smoother/faster convergence.

**Strengths:**

- Comprehensive empirical validation on a variety of vision tasks, showing clear positive benefit for large and small scale image classification, detection and segmentation
- (Somewhat less) comprehensive empirical validation on language tasks, showing improved performance on LRA and on 100m-param-scale language modeling
- Demonstrated benefit over mimetic initialization
- Addresses the ViT small-scale-data issue as well or better than mimetic initialization while being similarly simple
- Theoretically grounded
- Works for a variety of attention mechanisms

**Weaknesses:**

- Relatively small-scale language model evaluation (e.g., experiments requiring one GPU while training ViT-B presumably took multiple GPUs). But I still find the results convincing and promising. I know it's hard to do this on an academic budget. And the vision results alone justify my score.

**Questions:**

Did you try anything at the intersection of conditioned and mimetic initializations? For example, you could maybe pick a close orthogonal pair of matrices. Or you could use conditioned init for W_Q, W_K and mimetic init for W_V, W_O.

Did you try any ablations? (Like only initializing W_Q, W_K or only initializing W_V?)

Did you consider the output projection at all? It seems like it would be closely tied to W_V.

---

> ### Author Response · Authors · 2025-11-21
> **Response to Reviewer Reviewer 44Pd**
>
> We thank the reviewer for the time they have taken to review our paper. We answer all the points and questions they have raised in their review below. We have also updated our paper and added further sections and points in the paper that address the points/questions raised by the reviewer.
>
> **Reviewer 44Pd: Relatively small-scale language model evaluation (e.g., experiments requiring one GPU while training ViT-B presumably took multiple GPUs). But I still find the results convincing and promising. I know it's hard to do this on an academic budget. And the vision results alone justify my score.**
>
> We thank the reviewer for their understanding. Running larger-scale models is indeed difficult on an academic compute budget, and we were honest about this being a limitation in our Limitations Section 5 of the paper. However, we have trained a larger GPT-2 model (approximately 500 million parameters) on the WikiText-103 dataset and the TinyStories dataset. The tables below clearly show that our initialization achieves better performance, obtaining the lowest perplexity in all cases.
>
> **Perplexity of GPT-2 on WikiText-103**
>
> | Model                 | Perplexity |
> |-----------------------|------------|
> | Default               | 44.2       |
> | Mimetic               | 43.8       |
> | **Conditioned (ours)** | **42.7**   |
>
> **Perplexity of GPT-2 on TinyStories**
>
> | Model                 | Perplexity |
> |-----------------------|------------|
> | Default               | 2.39       |
> | Mimetic               | 2.32       |
> | **Conditioned (ours)** | **2.20**   |
>
>
> **Reviewer 44Pd: Did you try anything at the intersection of conditioned and mimetic initializations? For example, you could maybe pick a close orthogonal pair of matrices. Or you could use conditioned init for $W_Q$, $W_K$ and mimetic init for $W_V$, $W_O$.**
>
> Yes, originally we tried various combinations, thinking that since conditioned initialization introduces a conditioning bias and Mimetic introduces an attention-pattern bias (as Mimetic is obtained by mimicking converged attention patterns), combining the two might yield the best possible initialization. However, we found that combining them does not provide additional benefit compared to applying our conditioned initialization as designed. The table below shows what happens when we combine them in the way you suggest for Vision Transformers trained on ImageNet-1k:
>
> **ImageNet-1k Results (ViT family)**
>
> | Initialization                                      | ViT-B | DeiT-B | Swin-B | XCiT-M | DaViT-B |
> |-----------------------------------------------------|-------|--------|--------|--------|----------|
> | Default                                             | 80.3  | 81.6   | 83.4   | 82.6   | 84.3     |
> | Mimetic                                             | 80.5  | 81.6   | 83.5   | 82.6   | 84.4     |
> | Conditioned (our version)                           | **81.5** | **82.7** | **84.6** | **83.5** | **85.3** |
> | Conditioned ($W_Q$, $W_K$) + Mimetic ($W_V$, $W_O$) | 80.4  | 81.7   | 83.5   | 82.8   | 84.4     |
>
> For CIFAR10 and CIFAR100 using a ViT-Tiny, we observed similar behaviour:
>
> **CIFAR10 / CIFAR100 Results (ViT-Tiny)**
>
> | Initialization                                      | CIFAR10 | CIFAR100 |
> |-----------------------------------------------------|---------|-----------|
> | Default                                             | 92.4    | 71.7      |
> | Mimetic                                             | 93.6    | 75.0      |
> | Conditioned (our version)                           | **94.1** | **75.3**  |
> | Conditioned ($W_Q$, $W_K$) + Mimetic ($W_V$, $W_O$) | 93.5    | 75.1      |

---

> > ### Author Response · Authors · 2025-11-21
> >
> > **Reviewer 44Pd: Did you try any ablations? (Like only initializing $W_Q$, $W_K$ or only initializing $W_V$?)**
> >
> > Yes we tried many ablations that did various different initializations only on part of the $W_Q$, $W_K$, $W_V$ weights. Originally, we believed simply applying our initialization on $W_V$ should be enough as the rest of the attention block applies softmax which does a row normalization. However, we then realized that normalization is different to conditioning the Jacobian. Namely, you can have matrices with small norm but large condition number. This led us to the derivation of Theorem 3.2 which clearly shows that manipulating all three of $W_Q$, $W_K$ and $W_V$ would be the best approach. We have added a discussion on this in the appendix A.2 of the paper, see lines 916-924.
> >
> > **Reviewer 44Pd: Did you consider the output projection at all? It seems like it would be closely tied to $W_V$.**
> >
> > Yes we found that as long as we initialized $W_V$ in the way we did we could actually leave the initialization for $W_O$. If we also proceeded to initialize $W_O$ using our method we found it made no impact. We have added a discussion on this in the appendix in section A.2, see lines 926-931.

---

> > > ### Comment · Reviewer_44Pd · 2025-11-21
> > >
> > > Thanks for the thorough response. I'm happy to see the larger-scale language results.
> > >
> > > I've raised my score as I believe this should be highlighted at the conference.

---

### Official Review · Reviewer_sReQ · 2025-10-30

**Soundness:** 3
**Presentation:** 3
**Contribution:** 2
**Rating:** 6
**Confidence:** 3

**Summary:**

This paper proposes conditioned initialization, an initialization scheme for attention weights that explicitly targets the conditioning of the attention Jacobian. The authors present a theoretical analysis showing that the condition number of Jacobian can be upper-bounded in terms of the condition numbers of the query, key, and value matrices. Making these matrices well-conditioned at initialization is expected to stabilize optimization. Extensive experiments on various downstream tasks demonstrate that this initialization improves final accuracy and accelerates convergence compared to baseline methods on small models. The approach is simple to apply and does not require changes to existing training pipelines.

**Strengths:**

1. The paper provides a clear theoretical motivation and analysis, deriving an explicit upper bound related to attention optimization stability. This is both novel and well justified.
2. The paper is well written and easy to follow. The proposed method is straightforward to implement and architecture-agnostic.
3. Experimental results across various downstream tasks are promising. The method consistently improves performance and accelerates convergence.

**Weaknesses:**

1. The theoretical analysis optimizes an upper bound on the condition number of the Jacobian rather than the condition number itself. Although empirical results support the approach, the gap between the bound and the actual condition number is not fully characterized. It remains unclear whether a tighter bound would yield further improvements.
2. All experiments are conducted on relatively small models. It is unclear whether the benefits of conditioned initialization extend to large-scale models.
3. There is a typo on line 869: “The implementation of the ViTs were” should be “The implementation of the ViTs was.”

**Questions:**

1. Does conditioned initialization continue to provide performance gains and faster convergence when hyperparameters such as learning rate, warmup steps, and weight decay are re-tuned for each baseline? Could baseline methods catch up with light hyperparameter tuning?
2. Is the proposed method still effective for large-scale models?

---

> ### Author Response · Authors · 2025-11-21
> **Response to Reviewer sReQ**
>
> We thank the reviewer for the time they have taken to review our paper. We answer all the points and questions they have raised in their review below. We have also updated our paper and added further sections and points in the paper that address the points/questions raised by the reviewer.
>
> **Reviewer sReQ: The theoretical analysis optimizes an upper bound on the condition number of the Jacobian rather than the condition number itself. Although empirical results support the approach, the gap between the bound and the actual condition number is not fully characterized. It remains unclear whether a tighter bound would yield further improvements.**
>
> We thank the reviewer for their comment. We did honestly state in our limitations section on p. 9 of the paper that our approach only yields an upper bound on the condition number and thus serves as an indirect proxy for our initialization scheme and that developing methods that can efficiently estimate and control the exact condition number of the Jacobian would be a valuable direction for future work. Regardless our approach does do better than standard initializations within the literature on a variety of different applications and thus we feel is still useful for the community.
>
> **Reviewer sReQ: All experiments are conducted on relatively small models. It is unclear whether the benefits of conditioned initialization extend to large-scale models.**
>
> Our experiments focus on models in the million parameter range, where we consistently observe improvements in performance (see Section 4). While we did not evaluate billion-parameter models, we conducted experiments across a diverse set of tasks, including image classification, object detection, instance segmentation, language modeling, and long-range sequence modeling, which provides evidence that the proposed initialization is effective across different modalities and architectures. We also explicitly acknowledged in the Limitations section, see section 5 of main paper, that large-scale evaluation is an area for future work. Despite this constraint, the breadth of applications tested suggests that the method generalizes well beyond a single domain and based on this we believe is still useful to the community.
>
> **Reviewer sReQ: There is a typo on line 869: “The implementation of the ViTs were” should be “The implementation of the ViTs was.”**
>
> We thank the reviewer for pointing this out. We have corrected this typographical error and uploaded the new version of the paper.

---

> > ### Author Response · Authors · 2025-11-21
> >
> > **Reviewer sReQ: Does conditioned initialization continue to provide performance gains and faster convergence when hyperparameters such as learning rate, warmup steps, and weight decay are re-tuned for each baseline? Could baseline methods catch up with light hyperparameter tuning?**
> >
> > For the experiments we used the exact code bases provided in the papers of each baseline method, as mentioned on lines 870–871 in the original paper (now lines 936–942 in the newly uploaded version).
> > In each of these papers, the baseline models were already tuned using normalization techniques, learning-rate schedulers, warm starts, and weight decay. We did not change any of this; thus, the baselines we compare against have already been highly tuned.
> >
> > Conditioned initialization is a simple initialization scheme that is added on top of the baseline architectures, the training method does not change. This means anyway a model tuned, conditioned initialization can be added on top of that tuning as an initialization scheme. As an example, the ViT-B baseline hyperparameters (trained according to *“How to Train Your ViT? Data, Augmentation, and Regularization in Vision Transformers”* by Steiner et al.) that we used are as follows:
> >
> > **Table: Hyperparameter settings for ViT-B**
> >
> > | **Hyperparameter**       | **Value**    |
> > |--------------------------|--------------|
> > | batch size               | 1024         |
> > | epoch                    | 300          |
> > | learning rate            | 3.00E-03     |
> > | optimizer                | adamw        |
> > | weight decay             | 0.3          |
> > | label smoothing          | 0.1          |
> > | warmup epoch             | 20           |
> > | warmup learning rate     | 1.00E-05     |
> > | mixup                    | 0.8          |
> > | cutmix                   | 1            |
> > | drop path                | 0.1          |
> > | rand aug                 | 0.5        |

---

> ### Author Response · Authors · 2025-11-21
>
> **Reviewer sReQ: Is the proposed method still effective for large-scale models?**
>
> Unfortunately, we do not have the resources to run large-scale experiments in the billions-of-parameters range, as already stated honestly in the limitations section (Section 5) of the main paper. However, given the variety of tasks we evaluated in Section 4, and the consistent performance improvements obtained in each case, we believe our method is useful for the community.
>
> To further demonstrate our commitment, we additionally trained a larger GPT-2 model (approximately 500M parameters) on both the WikiText-103 dataset and the TinyStories dataset. The tables below show that our initialization achieves the lowest perplexity in all cases.
>
> **Perplexity of GPT-2 on WikiText-103**
>
> | Model                 | Perplexity |
> |-----------------------|------------|
> | Default               | 44.2       |
> | Mimetic               | 43.8       |
> | **Conditioned (ours)** | **42.7**   |
>
> **Perplexity of GPT-2 on TinyStories**
>
> | Model                 | Perplexity |
> |-----------------------|------------|
> | Default               | 2.39       |
> | Mimetic               | 2.32       |
> | **Conditioned (ours)** | **2.20**   |

---

> > ### Comment · Reviewer_sReQ · 2025-11-25
> > **Official responses by Reviewer sReQ**
> >
> > I have read the authors' rebuttal and other reviewers' comments. Thanks a lot to the authors for their sufficient extra experiments. I have no further comments and would keep my positive score.

---

### Official Review · Reviewer_ip1U · 2025-11-01

**Soundness:** 2
**Presentation:** 3
**Contribution:** 2
**Rating:** 4
**Confidence:** 4

**Summary:**

The paper presents a spectrum inspired approach to initialize the attention matrices to reduce the conditioning number of the attention Jacobian matrices with partial theoretical analysis. Extensive experiments demonstrate that using the presented initialization approach leads to consistently superior performance over the baselines.

**Strengths:**

+ The paper proposes a simple but effective intialization approach, which might have a better condition number of the Jacobian of the attention matrix with respect to the parameter matrices $W_Q$, $W_K$ and $Q_V$.
+ Extensive experiments demonstrate consistent improvements over the two baseline methods.

**Weaknesses:**

- The results in Lemma 3.1 and Theorem 3.1 cannot be listed as the (theoretical) contributions of the paper. These results are merely the formal expression of the gradients of a matrix function with respect to the parameter matrices via the Kronecker product. Similar results can be found in prior work, e.g., the appendix in [a].

- The results for Jacobian matrices developed in the paper might not complete due to ignorance of the stablization structure, e.g., Layer Norm (LN), RMSNorm, QKNorm. Thus, it is also questionable whether the upper bound of the condition number makes any sense in practical. If either LN, or RMSNorm, or QKNorm is introduced, can the proposed approach still yield improved performance comparing to the counterpart baseline methods?

- While the condition number of the proposed initialization strategy is reduced, it is merely a heuristic way to form the initialization for the attention matrices. Does it enable the training process stable? How about the effects of using the conditioned initialization on the learning curves?  In practice, when one of LN, or RMSNorm, or QKNorm or some combination of them is introduced, what aobut the learning curves?

- In previous work, there are many attempts to design stablized optimization algorithm for training Transformer. It would be more interesting if some evaluations to connect the proposed initialization strategy with the stablized algorithms. The reviewer is curious about that whether or not the proposed initialization still works when stablized optimization algorithms (or stablized structure, e.g., LN, RMSN, QKNorm, etc.) used.


[a] Taming Transformer Without Using Learning Rate Warmup, ICLR'25.

[b] Learning deep transformer models for machine translation. arXiv preprint arXiv:1906.01787, 2019.

[c] Query-key normalization for transformers. EMNLP 2020

[d] Scaling vision transformers to 22 billion parameters. ICML 2023.

[e] Root mean square layer normalization. NeurIPS 2019.

**Questions:**

- Please refer to the weaknesses.

---

> ### Author Response · Authors · 2025-11-21
> **Response to Reviewer  ip1U**
>
> We thank the reviewer for the time they have taken to review our paper. We answer all the points and questions they have raised in their review below. We have also updated our paper and added further sections and points in the paper that address the points/questions raised by the reviewer.
>
> **Reviewer  ip1U: The results in Lemma 3.1 and Theorem 3.1 cannot be listed as the (theoretical) contributions of the paper. These results are merely the formal expression of the gradients of a matrix function with respect to the parameter matrices via the Kronecker product. Similar results can be found in prior work, e.g., the appendix in [a].**
>
> We thank the reviewer for this comment. We would like to clarify that nowhere in the paper do we present Lemma 3.1 or Theorem 3.1 as standalone contributions. In the Introduction, we explicitly list our main contributions, the first of which states that "we establish a connection between the conditioning of self-attention Jacobians and the spectral structure of the query, key, and value matrices, motivating initialization schemes that explicitly target this property.'' This connection, linking Jacobian conditioning to the spectral structure of the query, key and value matrices is our actual contribution.
>
> Lemma 3.1 and Theorem 3.1 serve as important intermediate results that enable this connection, particularly through Theorem 3.2. To make their role as supporting results clearer, we have renamed Theorem 3.1 to Proposition 3.1. Please see the revised uploaded paper (note in the new uploaded paper as Theorem 3.1 has been renamed to Proposition 3.1, the counter makes Theorem 3.2 get named to Theorem 3.1 however we have not changed the statements in any way).
>
> We also thank the reviewer for directing us to reference [a]. We have carefully read this work, including Appendices A and B. While it contains useful derivations involving Kronecker-based Jacobian formulas, its scope differs from ours. Proposition 1 in [a] computes Jacobians for attention without softmax. Appendix A of [a] develops general identities for derivatives of matrix products, and Appendix B of [a] computes the Jacobian of self-attention with respect to the input matrix $X$. In contrast, our work derives the Jacobian with respect to the query, key, and value matrices and does so in the presence of softmax, which is required for our framework and subsequent initialization scheme. Thus, Lemma 3.1 and Proposition 3.1 remain essential components of our development. To acknowledge related efforts that compute Jacobians in specific cases using Kronecker products, we have now cited reference [a] in Appendix A.1, see lines 686-688.
>
> **Reviewer  ip1U: The results for Jacobian matrices developed in the paper might not complete due to ignorance of the stablization structure, e.g., Layer Norm (LN), RMSNorm, QKNorm. Thus, it is also questionable whether the upper bound of the condition number makes any sense in practical. If either LN, or RMSNorm, or QKNorm is introduced, can the proposed approach still yield improved performance comparing to the counterpart baseline methods?**
>
> We thank the reviewer for raising this point regarding normalization and stabilization mechanisms. We address the concern in two parts: (1) the relevance of our Jacobian analysis for vanilla self-attention, and (2) whether our conditioned initialization remains effective when modern normalization techniques (LN, RMSNorm, QKNorm) are applied.
>
> **(1) On the theoretical analysis for vanilla self-attention.**
> In the vanilla transformer formulation (e.g., *Attention Is All You Need*, Vaswani et al.), there is no normalization applied directly to the query, key, or value weights. The only modification is the fixed scaling factor $1/\sqrt{d}$ depending on the head dimension. As this factor does not depend on the parameters, differentiating the attention map simply scales the Jacobian. Since multiplying a matrix by a non-zero scalar does not change its condition number, this scaling cannot affect our conditioning analysis. Thus, our theoretical development for vanilla self-attention is correct and remains relevant as the motivation for our initialization scheme.

---

> > ### Author Response · Authors · 2025-11-21
> >
> > **(2) Applicability to modern attention mechanisms using LN, RMSNorm, or QKNorm.**
> > Modern architectures often apply additional normalization to the queries, keys, or values (as in RMSNorm or QKNorm, for example in Touvron et al.). In our submission, we explicitly noted this in the paragraph “Different forms of attention” (lines 257–262), where we state that our conditioned initialization can be applied on top of many newer forms of attention. We have now added a sentence to clearly state that our conditioned attention can be applied on top of those newer forms of attention that apply normalization to the parameters making up the attention matrix. We have updated the “Different forms of attention” paragraph to make this clear (see lines 260–263).
> >
> > We would also like to clarify a misunderstanding: we do not disable normalization when applying conditioned initialization. In all experiments, we retain the normalization layers used by each baseline architecture. That is, each model is trained exactly as described in the original paper, using the normalization they employ, and our initialization is applied **in addition to** the baseline architecture, not as a replacement. All baselines use the official implementations provided in their respective open-source repositories. For example, in Section 4, when we apply conditioned initialization to DeiT-B, the model’s QKNorm layers remain active, we do not remove them. This demonstrates that our method is naturally compatible with existing stabilization techniques.
> >
> > **(3) Normalization versus conditioning: fundamentally different purposes.**
> > Normalization techniques such as LN, RMSNorm, and QKNorm control the magnitude of weights or activations so that gradients do not explode or vanish. In contrast, conditioning concerns the *spectral structure* of the Jacobian. A matrix may have:
> >
> > - small norm but very poor conditioning, or
> > - large norm but excellent conditioning.
> >
> > For instance,
> >
> > $$A =
> > \left[
> > \begin{array}{cc}
> > 1 & 0 \\\\
> > 0 & 1
> > \end{array}
> > \right],
> > \quad
> > B =
> > \left[
> > \begin{array}{cc}
> > 10.1 & 0 \\\\
> > 0 & 10
> > \end{array}
> > \right]$$
> >
> > have very different Frobenius norms (matrix norm) but nearly identical condition numbers. Conversely,
> >
> > $$
> > C =
> > \left[
> > \begin{array}{cc}
> > 1 & 0 \\\\
> > 0 & 1
> > \end{array}
> > \right],
> > \quad
> > D =
> > \left[
> > \begin{array}{cc}
> > \sqrt{2} & 0 \\\\
> > 0 & 0.1
> > \end{array}
> > \right]
> > $$
> > have similar Frobenius norms but condition numbers $1$ and approximately $14.14$, respectively. Thus, normalization and conditioning address different aspects of stability. Our work targets the latter, **complementing** the effects of LN, RMSNorm, or QKNorm.
> >
> > **(4) Experimental evidence.**
> > For completeness, we conducted additional controlled experiments evaluating all architectures both with and without normalization applied to the inputs. Removing LN makes ViTs noticeably harder to train for all initialization schemes, yet our initialization consistently yields the best performance.
> >
> > **Table 1: Results without and with LayerNorm**
> >
> > | Initialization        | ViT-B | DeiT-B | Swin-B | XCiT-M | DaViT-B |
> > |-----------------------|-------|--------|--------|--------|----------|
> > | Original (no LN)      | 75.1  | 76.2   | 77.8   | 77.0   | 78.1     |
> > | Mimetic (no LN)       | 75.0  | 76.4   | 77.8   | 77.1   | 78.2     |
> > | **Ours (no LN)**      | **76.4** | **77.5** | **78.7** | **78.2** | **79.3** |
> > | Original (with LN)    | 80.3  | 81.6   | 83.4   | 82.6   | 84.3     |
> > | Mimetic (with LN)     | 80.5  | 81.6   | 83.5   | 82.6   | 84.3     |
> > | **Ours (with LN)**    | **81.5** | **82.7** | **84.6** | **83.5** | **85.3** |
> >
> > We further evaluated RMSNorm and QKNorm explicitly. In every configuration (no normalization, RMSNorm, QKNorm), our conditioned initialization improves performance relative to both the original initialization and Mimetic initialization.
> >
> > **Table 2: Results with RMSNorm and QKNorm**
> >
> > | Initialization          | ViT-B | DeiT-B | Swin-B | XCiT-M | DaViT-B |
> > |-------------------------|-------|--------|--------|--------|----------|
> > | Original                | 79.2  | 80.9   | 82.6   | 82.0   | 83.9     |
> > | Mimetic                 | 79.6  | 80.9   | 82.8   | 82.1   | 84.0     |
> > | **Ours**                | **81.0** | **81.9** | **83.7** | **83.0** | **84.6** |
> > | Original (RMSNorm)      | 80.2  | 81.4   | 83.2   | 82.1   | 84.0     |
> > | Mimetic (RMSNorm)       | 80.3  | 81.5   | 83.3   | 82.1   | 84.2     |
> > | **Ours (RMSNorm)**      | **81.3** | **82.4** | **84.3** | **83.3** | **85.1** |
> > | Original (QKNorm)       | 80.3  | 81.6   | 83.4   | 82.6   | 84.3     |
> > | Mimetic (QKNorm)        | 80.5  | 81.6   | 83.5   | 82.6   | 84.3     |
> > | **Ours (QKNorm)**       | **81.5** | **82.7** | **84.6** | **83.5** | **85.3** |
> >
> > All of this has been incorporated into a broader discussion in Appendix A.2 for the reader.

---

> ### Author Response · Authors · 2025-11-21
>
> **Reviewer ip1U: While the condition number of the proposed initialization strategy is reduced, it is merely a heuristic way to form the initialization for the attention matrices. Does it enable the training process stable? How about the effects of using the conditioned initialization on the learning curves? In practice, when one of LN, or RMSNorm, or QKNorm or some combination of them is introduced, what aobut the learning curves?**
>
> Conditioned initialization produces stable training across all architectures we evaluated. Instability during training typically manifests as failure to fit the data, divergence, or significantly degraded test accuracy. In all experiments, models initialized with the proposed method achieve higher final accuracy than both the default (truncated normal) and Mimetic initializations, indicating that the training dynamics remain stable and effective throughout.
>
> To make this observation explicit, we have added a new appendix section titled “Training Loss Curves” (see line 1001). This section includes the learning curves for every Vision Transformer variant under all three initialization schemes, when trained on ImagetNet-1k. The plots show that conditioned initialization exhibits smooth, stable optimization behaviour comparable to, and often better than, the baselines.
>
> As discussed in lines 257–262 of the main paper (“Different forms of attention”), conditioned initialization is designed to operate on top of modern attention variants that incorporate normalization mechanisms. In our implementation, these normalization layers are always retained, and conditioned initialization is applied in addition to them. Across all such settings, the learning curves remain stable and the final accuracy consistently improves, demonstrating that the method is compatible with, and complementary to, standard normalization techniques used in contemporary transformer architectures.

---

> > ### Author Response · Authors · 2025-11-21
> >
> > **Reviewer ip1U: In previous work, there are many attempts to design stablized optimization algorithm for training Transformer. It would be more interesting if some evaluations to connect the proposed initialization strategy with the stablized algorithms. The reviewer is curious about that whether or not the proposed initialization still works when stablized optimization algorithms (or stablized structure, e.g., LN, RMSN, QKNorm, etc.) used.**
> >
> > There exists a substantial body of work on stabilization techniques for training Transformers, including both optimization-level methods (such as adaptive clipping, learning-rate warmup, and stable Adam variants) and architectural stabilizers such as LN, RMSNorm, and QKNorm. These approaches primarily focus on controlling activation scales, gradient magnitudes, or update dynamics.
> >
> > The initialization strategy introduced in this paper addresses a different but complementary aspect of stability: the conditioning of the self-attention Jacobian. Whereas stabilized optimization algorithms aim to regulate the behaviour of gradients during training, our method improves the total singular-value structure of the attention layer at initialization.
> >
> > To examine whether the proposed initialization remains effective when stabilization mechanisms are present, we evaluate each baseline transformer in Section 4 using its original architecture, that includes stabilized normalization from the original papers. Conditioned initialization is applied on top of these components without modification. Across all architectures and normalization settings, the method continues to yield stable training and consistently higher accuracy than the default and Mimetic initializations. This demonstrates that the approach is compatible with existing stabilized optimization and architectural techniques.
> >
> > Finally, because initialization based conditioning influences only the starting point of training and does not alter the optimizer or the model structure, it integrates naturally with stabilized training algorithms.

---

### Meta-Review · Area_Chair_YfrV · 2026-01-06

**Summary:**

Overall the initial reviews were fairly positive (4,6,6,8) with many reviewers appreciating the simplicity, motivation, and effectiveness of the proposed method.

Reviewer ip1U notes that some of the theoretical results have appeared in prior work, raises questions about the relevance of the proposed method when normalization layers are added, along with asking whether the proposed method results in more stable optimization.

Reviewer sReQ and 44Pd are both largely positive in their reviews and ask if the benefits of the proposed approach can scale to larger models.

Reviewer jfhc points out that while the proposed initialization bounds the condition number of the Jacobian matrix it does not provide sufficient theoretical explanation for why this should improve optimization.

**Reviewer Concerns:**

The authors note that the theoretical results discussed by reviewer iP1U are not a core contribution of their work (rather an intermediate result in their derivation of the method) and have de-emphasized them in their revised manuscript (changing a Theorem to a Proposition) along with noting that the cited prior work, while similar, is derived for a different setting.

Likewise the authors respond to reviewer iP1U's questions on normalization and stability of optimization adequately in my view, pointing out that their experiments include normalization schemes mentioned by the reviewer and adding training loss curves, respectively.

In addition, the authors also include large-scale experiments, showing that their method also effectively scales to larger models.

Finally, the reviewers note that while their initialization scheme does not guarantee improved optimization on its own, the justification for bounding the Jacobian is provided in the literature.

Given these responses, I believe the authors have sufficiently addressed the bulk of the concerns raised by the reviewers.

**Reviewer Scores:**

The three reviewers that were initially positive in their reviews all note either maintaining their positive score or increasing their score post-rebuttal.

While the negative reviewer (ip1U) was not able to respond to the rebuttal before the discussion was closed, I am confident given the authors' responses and the support from the other reviewers that a consensus to accept the paper would be reached.

---

### Decision · Program_Chairs · 2026-01-26

Accept (Poster)